**Data Availability Statement:** All relevant data are within the manuscript and its Supporting Information files.

# Varietal susceptibility of maize to larger grain borer, *Prostephanus truncatus* (Horn) (Coleoptera; Bostrichidae), based on grain physicochemical parameters

Déthié Ngom[1,2]*, Marie-Laure Fauconnier[3], Paul Malumba[4], Cheikh Abdou Khadre Mbacké Dia[1], Cheikh Thiaw[2], Mbacké Sembène[1,5]

**1** Department of Animal Biology, Laboratory of Entomology and Acarology, Sciences and Technics Faculty, Cheikh Anta DIOP University, Dakar, Senegal, **2** Senegalese Institute of Agricultural Research (ISRA), Dakar, Senegal, **3** Agro-Biosystems Chemistry - Agronomy, Bioengineering and Chemistry (AgroBioChem), Gembloux Agro-Bio Tech, University of Liege, Gembloux, Belgium, **4** TERRA Teaching and Research Center, FoodIsLife Care, Gembloux Agro-Bio Tech, University of Liege, Gembloux, Belgium, **5** Biology Laboratory of Sahelo-Sudanese Animal Populations (BIOPASS) - Research Institute for Development (IRD), Dakar, Senegal

* dethie.ngom@ucad.edu.sn

## Abstract

Maize (*Zea mays* L) is one of main nutrients sources for humans and animals worldwide. In Africa, storage of maize ensures food resources availability throughout the year. However, it often suffers losses exceeding 20% due to insects such as the larger grain borer, *Prostephanus truncatus* (Horn) (Coleoptera; Bostrichidae), major pest of stored maize in the tropical countries. This study aims to select resistant varieties to reduce maize storage losses and explain the physicochemical parameters role in grains susceptibility. In the first study, maize grains were artificially infested under no-choice method with insects. Susceptibility parameters such as weight loss, grain damage, number of emerged insects, median development time and susceptibility index varied significantly through maize varieties. Dobie susceptibility index (SI) was assessed as a major indicator of resistance. The most resistant varieties were Early-Thaï, DMR-ES and Tzee-Yellow. Conversely, Synth-9243, Obatampa and Synth-C varieties were susceptible. SWAN, Across-Pool and Tzee-White were classified as moderately resistant varieties. The insect reproductive potential was significantly different in the nine maize varieties and Early-Thaï, DMR-ES and Tzee-Yellow varieties were the least favourable host. To assess the relationship between grains physicochemical characteristics and varietal susceptibility, moisture, total phenolics, palmitic acid, proteins, amylose, density and grain hardness were evaluated according to standardized methods. Palmitic acid, SI, insects emerged and grain damage were significantly and positively correlated with each other, and negatively correlated with grains hardness, phenolics and amylose contents. Maize susceptibility index was significantly and negatively correlated to amylose, and phenolics contents and positively correlated to palmitic acid content. This

**Funding:** Funding: This study has been totally funded by Cheikh Anta Diop University (grant number: 8729). The funder had no role in study design, data collection and analysis, decision to publish, or preparation of the manuscript.

**Competing interests:** The authors have declared that no competing interests exist.

study identified three resistant maize varieties to *P. tuncatus* and revealed that the major factors involved in this resistance were hardness, phenolic and amylose contents of grains.

## 1. Introduction

Ensuring global food security, nutrition and livelihood is one of the main challenges for the 21st century. Nowadays, maize (*Zea mays* L.), extensively grown in America, Asia and some parts of Africa, is the largest staple crop produced worldwide, which alone contributes over 20% of as food calories in parts of Africa and Mesoamerica [1]. In Sub-Saharan Africa, maize is widely consumed among many people and covers lean periods in some parts of these countries [2,3]. In these regions, production is generally seasonal while consumer needs extend throughout the year [4]. Thus, the storage of maize assures the food resources availability, which is one of the important factors of food security. However, huge amount of maize production is lost [5] between harvest and consumption. During storage, insects are the principal pests of maize [6] and one of the most important insects is the larger grain borer, *Prostephanus truncatus* (Horn) (Coleoptera; Bostrichidae) [7–9]. The infestation starts in the field and adults attack mainly whole or broken grains and flour during storage [10,11]. In developing countries, this insect is a serious pest of economic importance, causing maize-grain losses during storage ranging from 30% [12] to greater 40% of total production in 6 months [13]. This beetle reduces maize germination, increases the grain's moisture content [14,15] and facilitates the storage contamination by fungi and bacteria [16]. These fungi, particularly *Aspergillus flavus*, introduce a lot of aflatoxins in food products [10]. This carcinogenic substance poses many problems for consumers' health (consumed part) and for the export of African food products. *Prostephanus truncatus* was intercepted in Senegal in 2007 [17], and represents since then a threat to maize conservation in the country. The management actions taken against pests to reduce storage losses were primarily chemical. More recently, the fumigants ($CO_2$, $N_2$,. . .) and hermetical storage structures (walls, zinc, drums,. . .) are also used on *P. truncatus* control. However, residual insecticides currently used in grain storage are also subject to human health, environmental safety and pest resistance considerations due to misuse [18]. In developing countries, most of farmers have not access to hermetic storage structures and fumigants due to availability and cost reasons.

Use of varieties more tolerant to risk infestation during storage, is one of new principles and provisions on the integrated pest protection, encouraged by FAO [19] and European regulation to reduce insecticide use [20]. In fact, currently cultivated varieties, selected primarily for their high-yielding properties, often have increased susceptibility to insects, which would be due to the loss of their resistance characteristics to pest attack [21–23]. Resistant varieties are of particular interest in developing countries where lack of proper storage facilities can lead to substantial postharvest losses by insects. Several studies have concluded maize grains physicochemical constitution can play major role in damage during storage by insect pests [24–31]. Much research have claimed maize grains have properties that can moderate damage from *P. truncatus* infestations. Among them we have grain hardness [32,33], phenolic acids [34–37], proteins [38] and amylose content of total starch [39].

Thus, the knowledge of the physicochemical bases of the natural resistance, virtually unknown for maize varieties commonly used in Senegal, is crucial to the identification of resistance traits, which can be used for high-yielding insect-resistant varieties. Hence the present study has chosen the major insect pest of maize, *P. truncatus* in order to find solutions at entomological context of maize protection research in Senegal. The main objectives of present

study were (1) to evaluate the resistance of nine maize varieties to *P. truncatus* infestation in storage, and (2) to determine the physicochemical properties effect on the maize grains susceptibility.

## 2. Material and methods

### 2.1. Maize varieties used

Maize varieties not treated with insecticides used in these experiments were provided by Senegalese's seed services. Varieties evaluated were Early-Thaï, SWAN, DMR-ES, Tzee-White and Tzee-Yellow (obtained from Peanut and maize Seed Growers' Cooperative (COPROSA)-Nioro du Rip—Senegal) and Obatampa, Across-pool, Synth-C and Synth-9243 (obtained from National Center for Agricultural Research (CNRA)-ISRA /Bambey—Senegal). Maize grains were placed for three weeks in a freezer (at -5˚ C approximately) to eliminate any previous infestation before use.

### 2.2. Insect rearing

*Prostephanus truncatus* specimens used were obtained from the phytosanitary laboratory of Food Technology Institute (Senegal). Insects were reared in March 2017, at Entomology and Acarology laboratory of Sciences and Techniques Faculty (Cheikh Anta Diop University, Senegal). Glass jars (15 x 4 cm) were each loaded with 250 g of maize grains, and then 50 mixed sex adults were introduced into each jar. After 14 days, adults were separated to grains by sieving and sorting. Infested grains were incubated in insectarium under ambient temperature (25–35 ˚C) and relative humidity (70–80%) until newly adults emerged. Three generations were obtained from mass rearing techniques. Artificial infestation of samples was carried with young adults (two to three days-old) emerging from this rearing.

### 2.3. Screening of insect-maize varieties interactions

Varietal susceptibility was assessed through no-choice artificial infestation, which is better suited for screening candidate varieties [40]. In this experiment, three replications of 60 g of grains, placed in aerate glass jar with lid mesh (2 mm) were realized per maize variety. In each glass jar, three male/female pairs (two days-old) of *P. truncatus* were placed. The female stands out with the male by the high number of tubers on her head [41]. Adults were removed from the grains after 14 days laying period. Infested maize grains were incubated in insectarium at ambient temperature (25–35 ˚C) and relative humidity (70–80%) for 6 days. From there, emerged adults were counted daily until 55th day after the test start. Then, the following criteria were determined.

1. Total number of $F_1$ progeny emerged (TPE).

2. Multiplication rate per female: $Mr/♀ = \frac{TPE}{Number_{-female\ parents}}$

3. Net multiplication capacity: $Nmr = \frac{TPE}{Number_{-parents}}$

4. Median development time (MDT): is the period (days) from the middle of the oviposition period to the middle of the emergence (i.e. 50 percent of emergence) of the $F_1$ progeny [42].

5. Rate of increase per week: $Ri/w = \frac{Nmr\ X\ 7}{MDT}$

6. Dobie's Susceptibility Index (SI) to *P. truncatus* attacks, given by the formula:
   $SI = \frac{[ln(TPE)x\ 100]}{MDT}$

The Dobie susceptibility index, ranging from 0 to 11 [25,43], was used as the criterion to separate varieties into different resistance groups.

SI from 0 to 4: was classified as resistant varieties

SI from 4.1 to 7.0: was moderately resistant varieties

SI from 7.1 to 10.0: was susceptible varieties

SI ≥ 10.1: was classified as highly susceptible varieties.

7. Percent of grains attacked (grain damage percent): $\% \ Attack = \frac{Number_{-damaged \ grains}}{Number_{-total \ grains}} \ x \ 100$

8. Percent of grains weight loss was calculated using Boxal [44] counting and weighing method: $\% \ Weight \ loss = \frac{(B \ x \ E) - (C \ x \ D)}{E \ (B+C)} \ x \ 100$

B = number of damaged grains

C = number of undamaged grains

D = weight of damaged grains

E = weight of undamaged grains

## 2.4. Physicochemical analyses

The moisture content of maize grains was determined by drying in oven at 105 ˚C for 24 hours (method 967.03, [45]. Their protein content was determined through nitrogen determination by Dumas methodology using a Rapid N cube® combustion and analysis system (elementar, Nebraska, USA) (N × 6.25; method 981.10; [45]. Starch content was determined according to Ewers polarimetric method (ISO 10520: 1997) with a polarimeter (Bellingham Stanley Ltd. ADP220, UK). Amylose/amylopectin ratio of starches was evaluated with method of Morrison and Laignelet [46]. Crude oil content was extracted according to method of Folch et al. [47] using chloroform/methanol mixture (2/1, v/v). Fatty acids contents of extracted oil were determined after derivatization in BF3/methanol mixture by Gas Chromatography with Flame Ionization Detector (GC-FID). Total phenolics content was first extracted from 1 g of maize flour using pure methanol under sonication method. Methanolic extracts were assayed by adopting Bourgou et al. protocol [48] using colorimetric method with Folin-Ciocalteu reagent described by Singleton and Rossi [49]. Phenolics content were expressed as µg gallic acid equivalent per gram of maize dry matter (µg EAG / g DM). To evaluate hardness of grains, Stenvert hardness tester (Culatti microhammer mill, Labtech Essa, Belmont, Australia) fitted with a 2 mm aperture particles creen at a speed of 2500 rpm when empty was used on basis of method that described by Pomeranz et al. and Blandino et al. [50,51]. The time in seconds to collect 20 g of maize flour through 2 mm of mesh sieve was registered, at three replications for each maize variety. Average grain size parameters (length, width and thickness) were determined using a digital caliper (foot to slide). Grain density was determined from the ratio weight to volume measured on 30 individuals' grains according to Fox and Manley method [52]. Compositional characteristics were expressed in dry matter (DM). Each analysis was at least triplicated.

## 2.5. Statistical analyses

All susceptibility parameters as well as physicochemical characteristics were analysed using R software (R-3.4.1 version) [53]. Normality assumption was tested using Shapiro-Wilk's test and homogeneity of variance by Bartlett's test. For data whose series were followed normal distribution with homogeneous variance, one-way analysis of variance (ANOVA) was performed, then complemented with Tukey HSD test for separation of the mean values at α of 5%. Kruskal-

Wallis and Wilcoxon tests were used to analyze data whose series did not follow normal distribution and/or had not homogeneous variance (Moisture, Protein, Amylose, SI and Dmd).

Variables-PCA and Biplot-PCA: binary and multiple analyses with Spearman method were performed to determine interactions between variables. The choice of the factorial axe number for Principal Components Analysis (PCA) was carried according to elbow criterion which enabled us to obtain the maximum of inertia kept with the minimum factorial axes. The result suggest retain first two axes which respectively explain 46.8% and 20% of information, giving 66.8% of total inertia. Then, a Principal Components Analysis (PCA) of data was carried out in the purpose of highlighting the Principal Components (PC). Eleven (11) variables were considered PC of with contribution greater than 9.09% (average contribution, according to elbow criterion) for construction of axes considered. These were used to achieve a biplot combining the dispersion of PC (PCA) and maize varieties (Discriminant Analysis), in order to see the degree of correlation between the variables (susceptibility and physicochemical parameters) and the level of involvement of the physicochemical parameters to the maize varieties susceptibility.

## 3. Results

### 3.1. Total progeny emerged in $F_1$

For the adults emergence in $F_1$ (Fig 1), the results showed significant effects on the number of emerged insects due to maize varieties. The results have revealed that the group consisting Synth-9243, Obatampa and Synth-C varieties record between 7 to 9 times more insect than

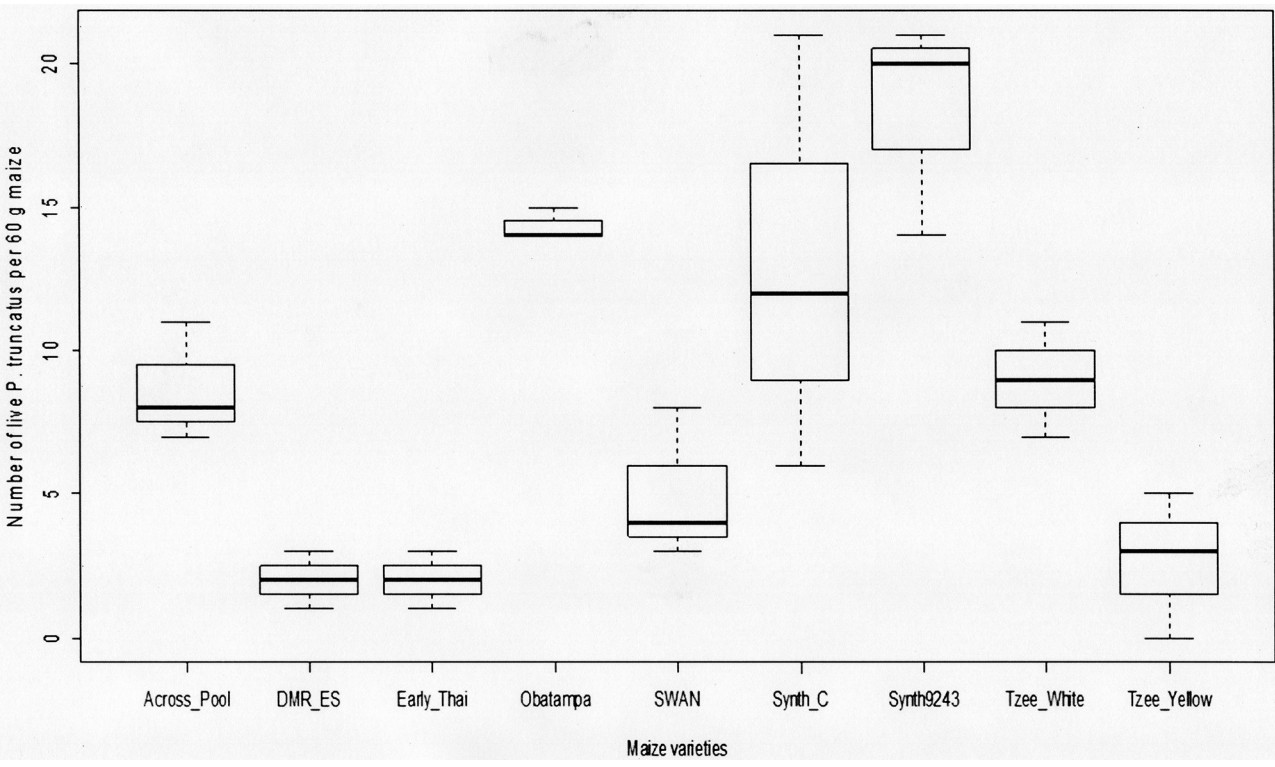

**Fig 1. Number of $F_1$ _P. truncatus_ progeny emerged on 60 g grains.** Number of $F_1$ _P. tuncatus_ progeny emerged on 60 g grains of each maize varieties after infestation out under no-choice method for 55 days on laboratory conditions. Overall differences of insects emerged between maize variety are significant at ANOVA test (P < 0.05).

group of Early-Thaï, DMR-ES and Tzee-Yellow varieties. Number of insects emerged from Synth-9243 variety was 9 times higher than that emerging from DMR-ES and Early-Thaï varieties, and 7 times more than that Tzee-Yellow variety. Tzee-White, Across-Pool and SWAN varieties had recorded a bit higher insects' number than the aforementioned group (DMR-ES, Early-Thaï and Tzee-Yellow varieties).

## 3.2. Reproductive dynamics of *P. truncatus* in $F_1$

The maize variety had also a significant effect (Table 1) on insects median development time (MDT), on rate of multiplication per female (Mr/♀) and on rate increase per week (Ri/w). Insects fed with Synth-9243, Synth-C and Obatampa varieties had the highest growth and femele rates per week, while the DMR-ES, Early-Thaï and Tzee-Yellow insects' varieties had the lowest. These last proved less favourable to insect development. In SWAN, DMR-ES and Across-Pool varieties, median development time of *P. truncatus* was 2 to 5 days longer than other varieties. The insect median development time in Tzee-White variety was only 28.74 ±0.26 days, widely shorter.

## 3.3. Susceptibility of varities to *P. truncatus* infestation

There were significant differences among varieties in percents of grains attacked and grains weight loss (Table 2). Early-Thaï, SWAN and Tzee-Yellow varieties had undergone lowest grains attacked and grains weight loss percentages, while Synth-9243 variety had recorded the highest percents. Significant differences (P = 0.017) were observed on susceptibility index (SI) between the maize varieties (Table 2). Out of the nine maize varieties evaluated for their *P. truncatus* resistance, three susceptibility groups were identified. The susceptible varieties group, with SI that revolves around 9, is composed of Synth-C, Obatampa and Synth-9243 varieties. These maize varieties had a SI 3 times higher than group of Early-Thaï, DMR-ES and Tzee-Yellow varieties (resistant varieties group). The other group (SWAN, Across-Pool and Tzee-White), categorized as moderately resistant, had a SI that ranged from 6.01±2.96 for SWAN to 6.94±3.37 for Tzee-White.

**Table 1. Effect of nine maize varieties on duration of development, number of emerged $F_1$ progeny of *P. truncatus*, multiplication rate per female, rate of increase per week and adults weight.**

| Varieties | TPE | MDT (days) | Mr/♀ | Ri/w |
|---|---|---|---|---|
| **Across-Pool** | 8.67±2.08[b] | 33.54±2.36[bc] | 2.89±0.69[c] | 0.31±0.09[c] |
| **DMR-ES** | 2.00±1.00[c] | 34.25±0.75[b] | 0.67±0.33[d] | 0.07±0.04[d] |
| **Early-Thaï** | 2.00±1.00[c] | 30.33±2.52[cd] | 0.67±0.33[d] | 0.08±0.04[d] |
| **Obatampa** | 14.33±0.58[a] | 31.64±1.74[c] | 4.78±0.19[b] | 0.53±0.04[b] |
| **SWAN** | 5.00±2.65[bc] | 37.00±1.00[a] | 1.67±0.58[c] | 0.16±0.09[d] |
| **Synth-9243** | 18.33±3.79[a] | 31.10±0.81[c] | 6.11±0.26[a] | 0.69±0.03[a] |
| **Synth-C** | 13.00±7.55[ba] | 31.42±0.08[c] | 4.33±0.52[b] | 0.48±0.08[bc] |
| **Tzee-White** | 9.00±2.00[b] | 28.74±0.26[d] | 3.00±0.67[c] | 0.37±0.08[c] |
| **Tzee-Yellow** | 2.67±2.52[c] | 31.73±0.07[c] | 0.89±0.54[d] | 0.10±0.09[d] |
| **Significativity** | *P < 0.05* | *P < 0.05* | *P < 0.05* | *P < 0.05* |
| | *$F_{8,18}$ = 9.976* | *$F_8$, chi² = 18.62* | *$F_{8,18}$ = 9.976* | *$F_{8,18}$ = 10.39* |

**TPE** = total progeny emerged; **MDT** = Median development time; **Mr/♀** = multiplication rate per female; **Ri/w** = rate of increase per week

P values of Tukey tests after ANOVA tests indicate the differences between means with the letters (a, b, c, d,. . .) at P < 0.05. Means followed by the same letter are not significantly different.

**Table 2. Parameters characteristic of maize sensitivity.**

| Varieties | % Attack | %Weight loss | SI | Resistance category |
|---|---|---|---|---|
| Across-Pool | 5.23±1.18[a] | 1.68±0.47[bc] | 6.05±2.68[b] | MR |
| DMR-ES | 5.12±0.64[a] | 1.77±0.20[b] | 3.05±1.09[c] | R |
| Early-Thaï | 1.10±0.38[e] | 1.10±0.38[c] | 2.85±0.78[c] | R |
| Obatampa | 4.00±0.43[b] | 1.92±0.06[b] | 9.74±0.52[a] | S |
| SWAN | 2.53±0.78[c] | 2.53±0.78[b] | 6.01±2.96[b] | MR |
| Synth-9243 | 6.59±1.71[a] | 3.80±0.56[a] | 9.72±0.17[a] | S |
| Synth-C | 4.42±2.33[abc] | 2.98±1.59[abc] | 9.76±0.24[a] | S |
| Tzee-White | 5.79±1.45[ab] | 3.19±0.65[ab] | 6.94±3.37[b] | MR |
| Tzee-Yellow | 2.59±1.63[ce] | 1.81±1.03[bc] | 3.75± 0.30[c] | R |
| Significativity | $P < 0.05$ | $P < 0.05$ | $P < 0.05$ | |
| | $F_{8,18} = 5.368$ | $F_{8,18} = 3.82$ | $F_8, chi^2 = 18.62$ | |

SI = susceptibility index; **MR** = moderately resistant. **S** = susceptible and **R** = Resistant.

**% Attack** = Percent attack of maize grains (damage percent); % **Weight loss** = Percent weight loss of maize grains. P values of Tukey tests after ANOVA tests indicate the differences between means with the letters (a, b, c,. . .) at P < 0.05. Means followed by the same letter are not significantly different.

## 3.4. Physicochemical parameters of maize grains

Biochemical and physical characteristics of maize varieties analysed are summarized in Table 3. There were significant differences among varieties for grain density (P = 0.031), grain hardness (P < 0.001), amylose (P = 0.003), initial moisture (P = 0.002), total phenolics (P < 0.001), palmitic acid (P < 0.001) and crude proteins (P = 0.004) grains contents. The highest protein content was observed with DMR-ES variety (13.69±0.42%), followed by Synth-C variety (13.57±0.05%), conversely the lowest protein content was registered in Early-Thaï variety (10.54±0.85%), followed by SWAN variety (10.64±0.11%). Both varieties, Early-Thai and SWAN, recorded the lowest palmitic acid content. The highest phenolics content (2864.16 ±121.00 and 2405.85±80.14 µg EAG / g DM) recorded in SWAN and Tzee-White varieties,

**Table 3. Physicochemical characteristics of maize varieties.**

| Varieties | Moisture (%) | TPP (µg EAG / g DM) | Palmitic (% DM) | Protein (% DM) | Amylose (% DM) | Density (g/mm³) | Hardness (s) (Crushing time) |
|---|---|---|---|---|---|---|---|
| Across-Pool | 11.15±0.15[bc] | 1461.60±120.55[e] | 17.71±0.06[b] | 12.77±0.69[ab] | 17.89±0.82[ab] | 1.43±0.16[cb] | 18.53±0.49[cb] |
| DMR-ES | 11.12±0.01[c] | 1297.17±74.64[e] | 17.50±0.10[c] | 13.69±0.42[a] | 18.57±0.71[a] | 1.40±0.16[c] | 17.90±0.52[cd] |
| Early-Thaï | 11.31±0.03[b] | 1650.23±40.26[d] | 16.91±0.03[f] | 10.54±0.85[d] | 18.25±0.49[a] | 1.46±0.17[ab] | 20.67±0.42[a] |
| Obatampa | 11.01±0.07[c] | 1111.94±69.50[f] | 17.84±0.06[a] | 12.04±0.25[bc] | 16.17±0.32[d] | 1.43±0.17[cb] | 14.90±0.38[e] |
| SWAN | 11.47±0.07[a] | 2864.16±121.00[a] | 16.48±0.04[g] | 10.64±0.11[d] | 17.62±0.1b[b] | 1.55±0.14[a] | 21.53±0.03[a] |
| Synth-9243 | 11.09±0.13[c] | 1159.28±40.16[f] | 17.34±0.06[d] | 11.93±0.32[bc] | 17.74±0.10[b] | 1.43±0.13[cb] | 19.23±0.17[b] |
| Synth-C | 10.97±0.06[cd] | 1234.68±78.66[ef] | 17.71±0.05[b] | 13.57±0.05[a] | 16.61±0.15[d] | 1.41±0.17[c] | 18.82±0.53[cb] |
| Tzee-White | 10.89±0.03[d] | 2405.85±80.14[b] | 17.41±0.04[d] | 11.93±0.06[bc] | 17.08±0.49[c] | 1.43±0.18[cb] | 16.97±0.33[d] |
| Tzee-Yellow | 11.52±0.00[a] | 1793.88±40.35[c] | 17.22±0.03[e] | 12.15±0.27[bc] | 17.11±0.01[c] | 1.45±0.15[cb] | 20.69±0.30[a] |
| Significativity | $P < 0.05$ | $P < 0.05$ | $P < 0.05$ | $P < 0.05$ | $P < 0.05$ | $P < 0.05$ | $P < 0.05$ |
| | $F_8, chi^2 = 24.07$ | $F_{8,18} = 173.2$ | $F_{8,18} = 261.8$ | $F_8, chi^2 = 22.54$ | $F_8, chi^2 = 23.47$ | $F_{8,261} = 2.159$ | $F_{8,18} = 85.32$ |

**Moisture** = Percentage of maize moisture content; **TPP** = Total phenolics content; **Palmitic** = Palmitic acid in fatty acids; **Protein** = Proteins content; **Amylose** = Amylose content in starch; **Density** = maize grains density; **Hardness** = maize grains hardness; **DM** = dry matter.

P values of Tukey tests after ANOVA tests indicate the differences between means with the letters (a, b, c, d, e,. . .) at P < 0.05. Means followed by the same letter are not significantly different.

were 2 times more than that those Synth-9243, Synth-C and Obatampa varieties. These last two varieties (Synth-c and Obatampa) had lowest amylose content, conversely to DMR-ES and Early-Thaï varieties, which had the highest amylose content in their grains total starch. The highest grains grinding time was recorded with SWAN variety (21.53±0.03 s.), followed by Tzee-Yellow (20.69±0.30 s.) then Early-Thaï (20.67±0.42 s.), revealing a high grain hardness, while Obatampa variety with grains faster crushed (14.90±0.38 s.), had the softest grains.

### 3.5. Binary correlations of evaluated variables

Spearman's correlation matrix showed significant binary correlations between physical, biochemical and susceptibility parameters of varieties (Table 4). The increase of grain hardness, amylose and phenolics contents in grains were negatively related with the varieties susceptibility index, while the increase of palmitic acid and protein contents were positively correlated with varieties susceptibility index. Number of emerged adults was negatively correlated to total phenolic content (rho = -0.528; $P_{(rho)}$ = 0.005), amylose content (rho = -0.513; $P_{(rho)}$ = 0.006), grains hardness (rho = -0.426; $P_{(rho)}$ = 0.027), but positively related to attack (rho = 0.616; $P_{(rho)}$ < 0.001), weight loss (rho = 0.654; $P_{(rho)}$ < 0.001), susceptibility index (rho = 0.846; $P_{(rho)}$ < 0.001) and palmilic acid (rho = 0.509; $P_{(rho)}$ < 0.001). Susceptibility index, positively related to palmitic acid content (rho = 0.475; $P_{(rho)}$ = 0.014), was inversely and signicantly correlated to amylose (rho = -0.673; $P_{(rho)}$ < 0.001) and phenolics content (rho = -0.451; $P_{(rho)}$ = 0.018), but no-signicantly correlated to grains hardness. Percent of grains attacked had negative significant relathionship with phenolics content (rho = -0.403; $P_{(rho)}$ = 0.037) and grains hardness (rho = -0.530; $P_{(rho)}$ = 0.004), but no-significant with amylose content. Percent of grains weight loss was negatively and significantly correlated to susceptibility index (rho = -0.530; $P_{(rho)}$ = 0.005), but no-significantly correlated to amylose, protein, palmitic acid and phenolics grains contents.

**Table 4. Spearman correlation matrix between physical, biochemical and varietal susceptibility parameters.**

| Variables | Moisture | TPP | Palmitic | Protein | Amylose | Density | SI | TPE | % Attack | % Weight loss | Hardness |
|---|---|---|---|---|---|---|---|---|---|---|---|
| **Moisture** | 1 | | | | | | | | | | |
| **TPP** | 0.430* | 1 | | | | | | | | | |
| **Palmitic** | -0.658*** | -0.697*** | 1 | | | | | | | | |
| **Protein** | -0.402* | -0.472* | 0.664*** | 1 | | | | | | | |
| **Amylose** | 0.346. | 0.190. | -0.428* | -0.082. | 1 | | | | | | |
| **Density** | 0.405* | 0.407* | -0.340. | -0.480* | -0.011. | 1 | | | | | |
| **SI** | -0.493** | -0.451* | 0.475* | 0.038. | -0.673*** | -0.022. | 1 | | | | |
| **TPE** | -0.491** | -0.528** | 0.509** | 0.038. | -0.513** | -0.111. | 0.846*** | 1 | | | |
| **%Attack** | -0.591** | -0.403* | 0.459* | 0.407* | -0.002. | -0.412* | 0.450* | 0.616*** | 1 | | |
| **%Weightloss** | -0.320. | -0.093. | -0.005. | -0.054. | -0.238. | -0.158. | 0.530** | 0.654*** | 0.681*** | 1 | |
| **Hardness** | 0.803*** | 0.487* | -0.799*** | -0.425* | 0.339. | 0.382* | -0.351. | -0.426* | -0.530** | -0.142. | 1 |

"*" Significant;

"**" Significant;

"***" Significant;

"." not significant **Moisture** = Percentage of maize moisture content; **TPP** = Total phenolics content; **Palmitic**: Palmitic acid in fatty acids; **Protein** = Proteins content; **Amylose** = Amylose content in starch; **Density** = maize grains density; **SI** = susceptibility index; **TPE** = total progeny emerged; **% Attack** = Percent attack of maize grains (damage percent); **% Weight loss** = Percent weight loss of maize grains; **Hardness** = maize grains hardness;

## 3.6. Multivariate analyses of parameters

The PCA graphic (Fig 2) revealed that the first axe (46.8%) and the second axe (20%) with 66.8% total inertia were best explained the Table 4 data. Principal components are colored (red to green on PCA) and selected according their significant contribution in two axes construction. Some of them participate significantly in first axe construction (SI: Cont_1 = 9.31%; Attack: Cont_1 = 10.33%; TPE: Cont_1 = 11.73%; Hardness: Cont_1 = 12.00%; Moisture: Cont_1 = 13.57% and Palmitic: Cont_1 = 14.51%) and others to second axe construction (TPE: Cont_2 = 10.15%; TPP: Cont_2 = 10.55%; Amylose: Cont_2 = 10.84%; Protein: Cont_2 = 12.55%; Density: Cont_2 = 12.83%; Weight loss: Cont_2 = 16.53% and SI: Cont_2 = 18.26%). A positive correlation were observed among the number of insects emerged in $F_1$ (TPE), percentage of

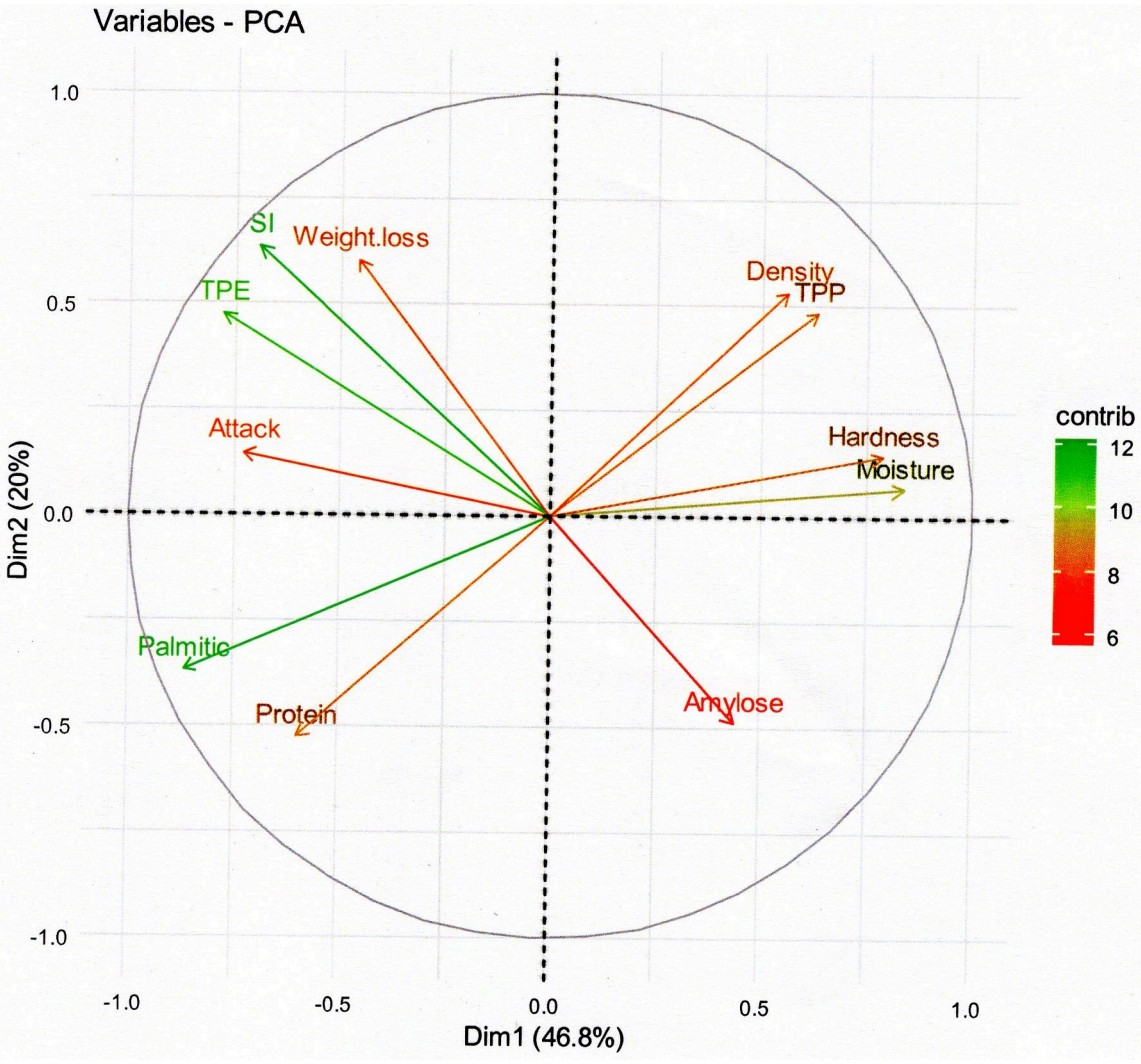

**Fig 2. Eigenvalue diagram on PCA of maize susceptibility parameters.** Eigenvalue diagram on Principal Component Analyses (PCA) of maize susceptibility parameters. The significance level for the contribution of the Principal Components (PC) is indicated by a color gradient. The annotation is **TPE**: Total Progeny Emerged, **SI**: Susceptibility Index, **Attack**: Attack percent of maize grains (damage percent), **Weight loss**: Weight loss percent of maize grains, **Moisture**: Percentage of grains moisture, **TPP**: Total Phenolics Content, **Palmitic**: Percentage of palmitic in grain fatty acids, **Protein**: Percentage of proteins in grain, **Amylose**: Percentage of amylose in grain starch, **Density**: Maize grains density, **Hardness**: Maize grains hardness.

attack (Attack), percentage of grains weight loss (Weight loss) and Susceptibility Index (SI), which are negatively correlated with total phenolics (TPP), moisture content and grains hardness.

The PCA-Biplot graphic (Fig 3) of varieties discrimination classified the nine maize varieties into three groups according their common and discriminants characteristics. Group_G1 (in green cycle) consists of the resistant varieties (DMR-ES, Early-Thai and Tzee-Yellow), Group_G2 (in yellow circles) composed of varieties with moderate resistance (Across-Pool,

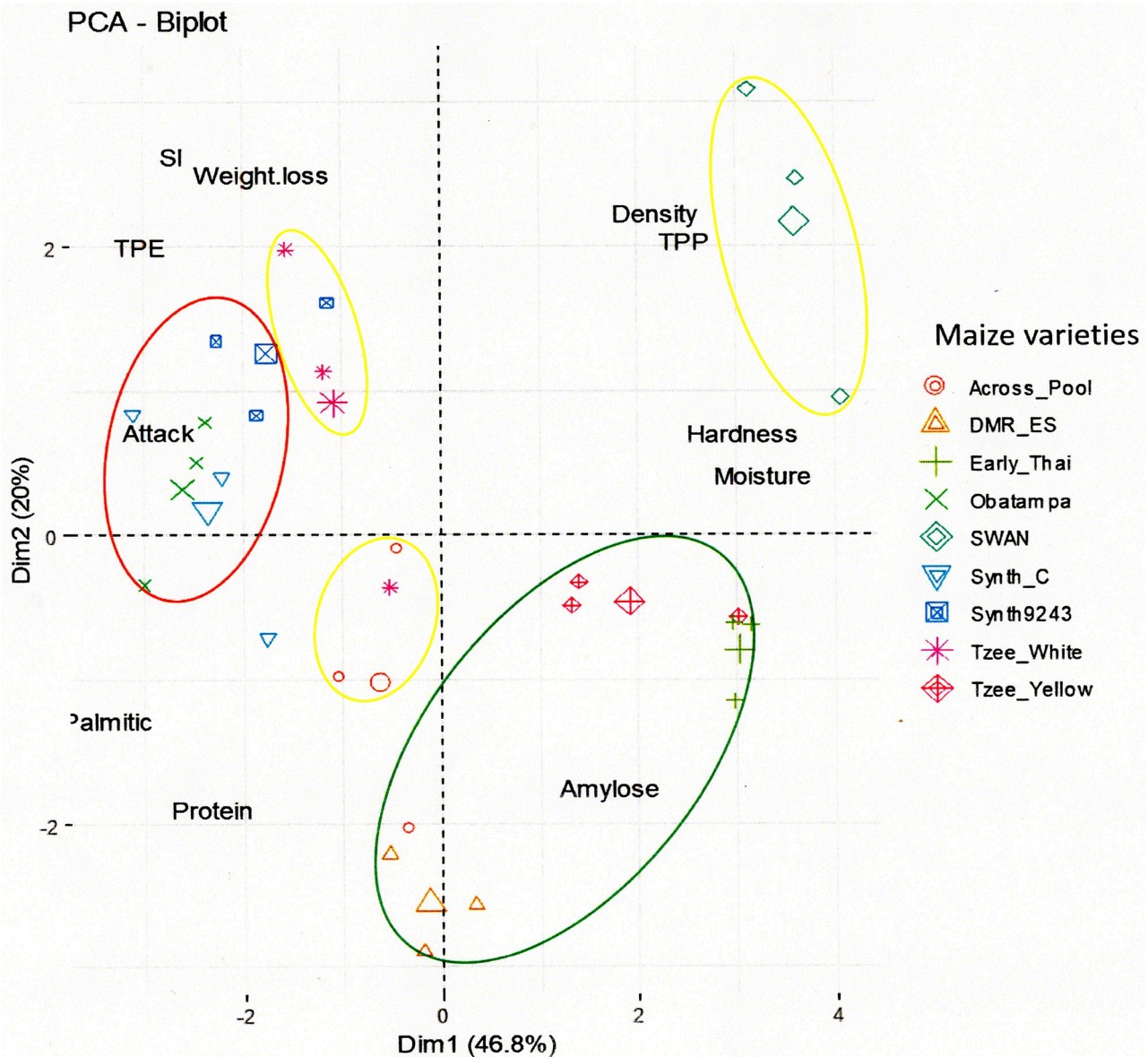

**Fig 3. Discrimination of maize varieties by physicochemical and susceptibility parameters (PCA-Biplot).** Discrimination of maize varieties (PCA-Biplot) by Discriminant Analysis (DA) on maize susceptibility parameters distribution. The annotation is **TPE**: Total Progeny Emerged, **SI**: Susceptibility Index, **Attack**: Attack percent of maize grains (damage percent), **Weight loss**: Weight loss percent of maize grains, **Moisture**: Percentage of grains moisture, **TPP**: Total Phenolics Content, **Palmitic**: Percentage of palmitic in grains fatty acids, **Protein**: Percentage of proteins in grains, **Amylose**: Percentage of amylose in grains starch, **Density**: maize grains density, **Hardness**: maize grains hardness. The susceptibility level of varieties is classified on three groups and indicated by a color gradient; Group_G1: resistant varieties (green circle), Group_G2: moderately resistant varieties (yellow circles) and Group_G3: susceptible varieties (red circle).

SWAN and Tzee- White) and Group_G3 (in red circle) with susceptible varieties (Synth-C, Synth-9243 and Obatampa) against *P. truncatus*.

## 4. Discussion

### 4.1. Susceptibility parameters

The susceptibility characteristics of maize grains exposed to *P. truncatus* for 55 days in the laboratory, were highlighted in this study. First emergence of adults has shown all nine varieties are favourable to the insect development. However, variety factor had a significant effect on susceptibility parameters such as number of emerged insects, median development time, percent of grains attacked, percent of grains weight loss and susceptibility index. The current study indicate also that the number of emerged adults is positively related to attacks and weight loss percents of grains, which is consistent with previous work [22,54]. Early-Thaï, SWAN and Tzee-Yellow varieties that had the lowest susceptibility index (resistant varieties) have recorded few insects emergence in $F_1$ and exhibited lowest percents of attack and weight loss of grains. Conversely, Synth-9243, Synth-C and Obatampa varieties that had susceptibility index threefold higher (susceptible varieties), and recorded the highest insects emergence with rates of insects and females multiplication per week significantly higher. Attack and weight loss levels of grains, which resulted from both adult and larval feeding activities, were high in this maize varieties. Whereas the reverse is observed in resistant varieties, whose insect reproduction was negatively affected and the damage on the grains was minimal. These significant variations in the susceptibility parameters of the maize varieties show the natural capacity of particular varieties to resist against *P. truncatus* infestation. Such sources of maize resistance may be explained by antibiosis and antixenosis mechanisms on the basis of the grains physicochemical characteristics.

### 4.2. Relation of physicochemical parameters to maize susceptibility

Past studies on maize susceptibility to *P. truncatus* (which mentioned in introduction section) have reported an essential role of grains physicochemical parameters. Our study showed the maize susceptibility to the larger grain borer could be explained by physical mechanisms (hardness, density,...) and/or biochemical mechanisms such amylose, moisture, total phenolics, palmitic acid and crude protein contents, which have proved very different in our varieties. The PCA-Biplot graphic (Fig 3) of varieties distribution according their discriminants parameters (main components), explain the varieties classification by different grain characteristics.

   **Group_G1**: (DMR-ES, Early-Thai and Tzee-Yellow, resistant varieties).

   On PCA-Biplot graphic, these three varieties are particularly characterized by high levels of amylose and moisture contents and high grains hardness. As stated earlier (Table 4), these parameters were inversely correlated to grains susceptibility index, which reveals that they are involved and seem to be the principals' factors for the resistance of these maize varieties. In fact, increasing of amylose content can negatively affect the insect reproduction, interfering on digestion mechanism (antibiosis action). It is known that high amylomaize are less susceptible to *P. truncatus* larval feeding activities, due to their low digestive alpha-amylase [39]. Otherwise, high amylose content was reported to reduces the damage in maize stored and amyloidosis is detrimental to weevil larval survival [24]. However, the no-significant correlation of amylose content with percentages of grains attacked and weight loss (Table 4), suggests that high grains amylose content not appear to have anti-feedant effect on adults. This is in agreement with previous studies [39], who had found high digestive alpha-amylase in *P. truncatus* adults. Furthermore, the moisture content increases inversely with all maize susceptibility

parameters (Table 4), so, its high content in the grain less favors the insects' development, as indicated Arnason et al. [55]. Grain hardness, previously considered as indicator of maize resistance against insects [36] would act as mechanical strength (antixenosis action) for these maize varieties, to reduce the insects feeding and oviposition. The hardness, which increases with amylose content [38], was reported as one of maize resistance source against *P. truncatus* [22,56].

As a result, the combination of this both resistance mechanisms (antibiosis and antixenosis) in these three maize varieties, involve complex interactions manifested as grains modifications which lead to limited grain accessibility (physical barriers) and creates toxicity for insects [12].

**Group_G2**: (SWAN, Across-Pool and Tzee- White, moderate resistant varieties).

These three moderate resistant maize varieties showed in specific characteristics for each of them.

Across-Pool variety is characterized by its palmitic acid, protein and amylose contents, constituents that have antagonistic actions against the *P. truncatus* infestation. Indeed, this work showed that the protein content increases with percent of grains attacked. This could mean that susceptibility is dependent on the nutritional content of the maize varieties studied. Our result was in agreement with previous studies, who found that protein content increased with maize susceptibility to maize weevil, *Sitophilus zeamais* M. [31] and with maize grains damage caused by Angouillois grain moth, *Sitotroga cerealella* O. [28]. In contrast, other studies reported also maize grain hardness was positively correlated to zein family of proteins [32,33,38]. The importance role of the proteins content in this study suggest a more detailed analysis to quantify proteins families in our varieties and their involved in the larger grain borer feeding and oviposition. Although the proteins are favourable to insect development, but the high amylose content is asset for maize resistance by impacting negatively the insects' digestion and probably caused a mortality of some larvae.

SWAN variety is characterized by high phenolics content and very hard grains. The maize resistance mechanism by phenolics content (ferulic and p-coumaric acids) is related one part to grains hardness (structural barriers), which is clearly dependent on presence of these two phenolic acids in grains pericarp and aleurone layer. Other part, its mechanism is associated to anti-feedants and toxic properties of free phenolic acids [26,57], that cause damage to midgut cells of the insects [56]. It was reported also that phenolic amines, localized in the aleurone layer [57], inhibit glutamate-dependent neuron receptors in insects [58]. However, ferulic acid, which largely composed phenolic acids in maize grains [37,59], has moderate nuisance on *P. truncatus* [35]. SWAN variety has an antibiosis and antixenosis actions based on grains hardness (physical barriers) and biochemical nuisances (anti-feedants and toxic free form), which are moderately effective against *P. truncatus* feeding and oviposition.

Tzee-White variety has soft grains with high phenolics content and low moisture content. This significant amount of phenolics content is expected to make it resistant against *P. truncatus*, however the low moisture content and low hardness of its grains would have favored the insect ovipostion, reduced greatly the median development time and allows the insect to minimize larval mortality due to phenolics.

SWAN and Tzee-White varieties, although they had phenolics content widely higher than whose DMR-ES, Early-Thai and Tzee-Yellow varieties, their susceptibility index was 2 times lower than that of these varieties. This shows, even though phenolics content is considered as good indicator of resistance, alone can not protect maize grains against *P. truncatus*.

**Group_G3**: (Synth-C, Synth-9243 and Obatampa, susceptible varieties).

These three susceptible maize varieties showed in common high palmitic acid content, low amylose content and specific characteristics for each of them. Their vulnerability would be due particularly to low amylose content and high palmitic acid content, which were positively

correlated to all grains susceptibility parameters (Table 4). Indeed, earlier studies reported that fatty acids, particularly palmitic acid, attract insects [60] and represent a considerable nutritional and functional value for them [61]. However, the effect of palmitic acid on *P. truncatus* oviposition and larvae-feed needs further investigation. For Obatampa, furthermore the high palmitic acid content, this variety consists softer grains, making it vulnerable and more favourable for insect development than Synth-C variety.

In summary, interesting varieties for maize resistance to postharvest insects, in particular to *P. truncatus* were identified in nine maize varieties selected for farmers under artificial infestation by the larger grain borer. Resistance was related more to grains amylose content, followed the grains hardness than elevated levels of phenolics content.

## 5. Conclusion

In conclusion, we can say our study showed host variety resistance in maize production can be a useful component of integrated pest management of *P. truncatus*. Results obtained illustrate that all maize varities can be attacked by the larger grain borer, which was found to exert different degrees of damage on varieties. Significant differences in susceptibility to *P. truncatus* were observed among nine maize varieties evaluated. Most resistant varieties were Early-Thaï, DMR-ES and Tzee-Yellow, followed SWAN, Across-Pool and Tzee-White (moderately resistant varieties) then the most susceptible varieties what are Synth-C, Synt-9243 and Obatampa. Physical and biochemical parameters associated to the maize resistance were more amylose content, followed the grains hardness than phenolics content. The most resistant maize against *P. truncatus* can be described as a variety with increased amylose and phenolics contents and hard grains. Thus, Early-Thaï, DMR-ES and Tzee-Yellow varieties presenting good nutritional and entomological profile may be potential maize varieties to be used to reduce the maize postharvest loss due to *P. truncatus* in tropical countries. Further studies on maize attractiveness in relation of color parameters of grain (L*, a*, b*, ΔE, ΔC and H) and volatile organic compounds of maize varieties against the larger grain borer is also being performed. Beyond this study in storage pests and characteristics of resistance, an ideal breeding programme would consider yield as well as pests and diseases in field.

## Acknowledgments

We would like to thank Mr. Issa CISSE (COPROSA / Nioro-Senegal) and Dr. Moustapha GUEYE (CNRA-ISRA Seed Service / Senegal) for their collaboration in collecting maize samples. We gratefully acknowledge Gembloux Agro Bio-Tech Faculty of Liège University (Belgium), particularly General and Organic Chemistry laboratory staff (Mr Danny TRISMAN, Mr Thomas BERTRAND,. . .) and TERRA Teaching and Research Center staff (Vanessa ARDITO, Nathalie, Romain,. . .) for their unfailing support on maize grains physicochemical characterization.

## Author Contributions

**Conceptualization:** Déthié Ngom, Cheikh Thiaw, Mbacké Sembène.

**Data curation:** Déthié Ngom.

**Formal analysis:** Déthié Ngom, Marie-Laure Fauconnier, Paul Malumba.

**Funding acquisition:** Déthié Ngom, Mbacké Sembène.

**Methodology:** Déthié Ngom, Paul Malumba, Cheikh Thiaw.

**Resources:** Marie-Laure Fauconnier, Paul Malumba, Mbacké Sembène.

**Software:** Déthié Ngom, Cheikh Abdou Khadre Mbacké Dia.

**Validation:** Déthié Ngom, Mbacké Sembène.

**Writing – original draft:** Déthié Ngom, Cheikh Thiaw, Mbacké Sembène.

**Writing – review & editing:** Déthié Ngom, Marie-Laure Fauconnier, Paul Malumba.

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
