## [Decision Letter · Decision Letter 0]

19 Feb 2020

PONE-D-19-25206

Varietal susceptibility of maize to larger grain borer, Prostephanus truncatus (Horn) (Coleoptera : Bostrichidae), based on grain physicochemical parameters

PLOS ONE

Dear Mr., NGOM,

Thank you for submitting your manuscript to PLOS ONE. After careful consideration, we feel that it has merit but does not fully meet PLOS ONE’s publication criteria as it currently stands. Therefore, we invite you to submit a revised version of the manuscript that addresses the points raised during the review process.

The reviews were generally positive towards publication after making some improvements.  It is very important that you pay attention to the suggestions about improving the statistical analysis of the data. You may also want to have the manuscript reviewed for grammar usage to improve its flow and readability.

We would appreciate receiving your revised manuscript by Apr 04 2020 11:59PM. To enhance the reproducibility of your results, we recommend that if applicable you deposit your laboratory protocols in protocols.io, where a protocol can be assigned its own identifier (DOI) such that it can be cited independently in the future. For instructions see: http://journals.plos.org/plosone/s/submission-guidelines#loc-laboratory-protocols

We look forward to receiving your revised manuscript.

Kind regards,

Craig Eliot Coleman, PhD

Academic Editor

PLOS ONE

Journal Requirements:

Reviewers' comments:

Reviewer's Responses to Questions

**Comments to the Author**

1. Is the manuscript technically sound, and do the data support the conclusions?

Reviewer #1: Partly

Reviewer #2: Yes

2. Has the statistical analysis been performed appropriately and rigorously? 

Reviewer #1: No

Reviewer #2: Yes

3. Have the authors made all data underlying the findings in their manuscript fully available?

Reviewer #1: Yes

Reviewer #2: Yes

4. Is the manuscript presented in an intelligible fashion and written in standard English?

Reviewer #1: Yes

Reviewer #2: Yes

5. Review Comments to the Author

Reviewer #1: The paper examined susceptibility of nine varieties to larger grain borer infestation. The premise for the work is valid, and methods followed were perfect. However, the authors' Tables 1-3 have errors. Under Results they should provide one way ANOVA results for each variable studied like F value, df (numerator and denominator) and P value. The mean separation test using Tukey's was wrongly interpreted by the authors. The authors need to consult a statistician for these results and rerun the analysis. For example in all Tables the highest value should get letter 'a', but in their results for all variables teh lowest values was given 'a'. The correlations should be confirmed by checking scatter plots to see if there are any spurious correlations (two sets of data points where a straight line will go through it). The PCA plots need to be very clear.

The total progeny produced is very low for 3 males and females? What was the mortality of the adults in the 14 days before they were removed? I would have expected over 100 adult progeny emerging?

Other minor changes:

In abstract and elsewhere, choice not choise

Don't show stats in abstract.

Line 51. grains should be grain's

Line 57. Add period after chemical. Then start a new sentence: More recently the fumigant ________ and hermetic storage structures such as -------- are other methods of P. truncatus control.

Line 73. .......high-yielding....

Line 77-78. Delete this long sentence as it deals with results.

Line 90. space after 250 g

Line 98. space after 60 and g

Line 99. space after 2 and mm.

What was the moisture of varieties at start of test?

Why have 4 levels of significance. Use P less than P < 0.05 or give actual P-value. Delete lines 153-158 and remove several levels of significance from table footnotes.

Significance is always P < 0.05 not P > 0.01.

Other than a few issues cited above the manuscript is a valuable contribution.

Reviewer #2: Manuscript Review:

Title: Variation in varietal susceptibility of maize to larger grain borer (LGB), Prostephanus truncates (Horn) (Coleoptera: Bostrichidae), in relation to corn kernels physiocochemical parameters

Plos one

Authors: NGOM Dethie1,2, FAUCONNIER Marie-Laure3, MALUMBA Paul Kamba4, DIA CHEIKH Abdou KHADRE Mbacke1, THIAW Cheikh2, SEMBENE Mbacke1,5.

Laboratory of Entomology and Acarology, Department of Animal Biology, Sciences and Technics Faculty, Cheikh Anta DIOP University, PO Box 5005 Dakar, Senegal. Dethie.ngom@ucad.edu.sn

Senegalese Institue of Agricultural Research (ISRA), PO Box 3120, Dakar, Senegal.

Agro-Biosystems Chemistry – Agronomy, Bioengineering and Chemistry (AgroBioChem) – Gembloux Agro-Bio Tech – University of Liege, Deportees Passage, 2-5030 Gembloux – Belgium.

FoodIsLife Care, TERRA Teaching and Research Center – Gembloux Agro-Bio Tech – University of Liege, Deportees Passage, 2-5030 Gembloux – Belgium.

Biology Laboratory of Sahelo-Sudanese Animal Populations (BIOPASS) – Research Instititue for Development (IRD), Bel0Air PO Box 1386, Dakar, Senegal

Summary of the research:

The authors investigated several different maize varieties for resistance to the larger grain borer. They looked at weight loss, grain damage, number of emerged insects, median development time and susceptibility index. They did this using a no-choice test, where corn kernels of each variety were artificially infested and then evaluated for the previously mentioned parameters. They also looked to link physiochemical characteristics of the grain to varietal susceptibility by looking at grain moisture, total phenolics, palmitic acid, proteins, amylose, density, and grain hardness. The significance of this work is that farmers will have a better understanding of the varieties available and their ability to provide protection of the crop even after harvest. The study was able to identify 3 resistant varieties for potential use.

Methods:

Maize Varieties Used

Evaluated the varieties: Early-Thai, SWAN, DMR-ES, Tzee-White and Tzee-Yellow, Obatampa, Across-pool, Synth-C and Synth-9243.

Physiochemical Analysis

Moisture content was determined by drying, protein content was determined through nitrogen determination by Dumas methodology, starch content was determined using Ewers polarimetric method, amylose/amylopectin ratio of starches was evaluated with Morrison and Laignelet method, crude oil content was extracted via Folch et al., methodology, fatty acid content was determined after derivatization in BF3/Methanol mixture by gas chromatography, and total phenolics was also analyzed. They also evaluated grain hardness using the Stenvert hardness test, kernel density was determined from the ratio of grain weight to volume measured using Fox and Manley method. They had 3 replicates for each analysis.

Screening of insect-maize varieties interactions

Varietal susceptibility was assessed through no-choice artificial infestation. Three Male/Female pairs were placed into the jars of maize and the adults were removed after 14 days. The jars of infested kernels were kept in an insect chamber until 20 days post infestation. Emerged adults were then counted weekly until 55 days post infestation. They had 3 replicates for each maize variety.

Sample Size:

The authors made a point to emphasis the fact that they replicated their results at least 3 times for each important step.

Analysis seemed appropriate:

Analysis appears to be appropriate for the work presented. The methodology appears sound. At every step the authors made a point to site previous work which would back up their methodology. This is a very basic and already proven experimental design as far as testing for resistant varieties of grain.

Comments:

-Lines 16-36 The abstract seems too long and lacking a good introductory statement to pull the reader in, and a similarly strong concluding statement. I would not put all these results and numbers in the abstract, give brief and clear statements about the findings. I would like to see these added, as many folks read the abstract first to decide:

1. What the paper is about and

2. Will they read all of it

-I was happy to see the results clearly separated from the discussion, which made it easier to evaluate.

-Rationale for the work was strong!

-I like the fact that the authors provide citations to strengthen their methods at every step and show that their methods have a strong background in the literature.

-I was pleased to see the authors went variety by variety and explained what they found in their analysis in detail in the discussion.

-Overall, considering the current literature published on the resistant varieties of maize to the larger grain borer, I think this paper is a needed contribution to the current body literature on the subject. There seem to be very few studies that look resistant varieties of maize for this species and many people in general are quite biased against host plant resistance in stored products. However, considering invasive and devastating pests like the larger grain borer, this strategy could be quite useful.

I recommend accept with minor revisions

6. PLOS authors have the option to publish the peer review history of their article (what does this mean?). If published, this will include your full peer review and any attached files.

Reviewer #1: No

Reviewer #2: No

---

## [Author Response · Author response to Decision Letter 0]

23 Mar 2020

Reviewer #1: 

The paper examined susceptibility of nine varieties to larger grain borer infestation. The premise for the work is valid, and methods followed were perfect. However, the authors' Tables 1-3 have errors. Under Results they should provide one way ANOVA results for each variable studied like F value, df (numerator and denominator) and P value. The mean separation test using Tukey's was wrongly interpreted by the authors. The authors need to consult a statistician for these results and rerun the analysis. For example in all Tables the highest value should get letter 'a', but in their results for all variables teh lowest values was given 'a'. The correlations should be confirmed by checking scatter plots to see if there are any spurious correlations (two sets of data points where a straight line will go through it). The PCA plots need to be very clear.

Done

The total progeny produced is very low for 3 males and females? What was the mortality of the adults in the 14 days before they were removed? I would have expected over 100 adult progeny emerging? 

The low progeny produced by the three male/female pairs can be explained by the small quantity of grains used (60 g of grains only). Indeed, 18 emerged insects (Synth-9243 variety for example) in only 60 g of grains for first offspring (F1), represents high reproductive potential for this voracious insect on stored maize.

There had not mortality of adults during the 14 days oviposition. This would be explained by the fact that we used young adults (two days-old).

Other minor changes:

In abstract and elsewhere, choice not choise: Done

Don't show stats in abstract: Done

Line 51. grains should be grain's : Done

Line 57. Add period after chemical. Then start a new sentence: More recently the fumigant ________ and hermetic storage structures such as -------- are other methods of P. truncatus control: Done

Line 73. .......high-yielding.... : Done

Line 77-78. Delete this long sentence as it deals with results: Done

Line 90. space after 250 g : Done

Line 98. space after 60 and g : Done

Line 99. space after 2 and mm. : Done

What was the moisture of varieties at start of test? At start of test, the moisture content of varieties was between 10.89±0.03 % (Tzee-White variety) and 11.52±0.00 % (Tzee-Yellow variety)

Why have 4 levels of significance. Use P less than P < 0.05 or give actual P-value. Delete lines 153-158 and remove several levels of significance from table footnotes: Done

Significance is always P < 0.05 not P > 0.01. Done

Reviewer #2: 

Manuscript Review:

Title: Variation in varietal susceptibility of maize to larger grain borer (LGB), Prostephanus truncates (Horn) (Coleoptera: Bostrichidae), in relation to corn kernels physiocochemical parameters

Plos one

Authors: NGOM Dethie1,2, FAUCONNIER Marie-Laure3, MALUMBA Paul Kamba4, DIA CHEIKH Abdou KHADRE Mbacke1, THIAW Cheikh2, SEMBENE Mbacke1,5.

Laboratory of Entomology and Acarology, Department of Animal Biology, Sciences and Technics Faculty, Cheikh Anta DIOP University, PO Box 5005 Dakar, Senegal. dethie.ngom@ucad.edu.sn

Senegalese Institue of Agricultural Research (ISRA), PO Box 3120, Dakar, Senegal.

Agro-Biosystems Chemistry – Agronomy, Bioengineering and Chemistry (AgroBioChem) – Gembloux Agro-Bio Tech – University of Liege, Deportees Passage, 2-5030 Gembloux – Belgium.

FoodIsLife Care, TERRA Teaching and Research Center – Gembloux Agro-Bio Tech – University of Liege, Deportees Passage, 2-5030 Gembloux – Belgium.

Biology Laboratory of Sahelo-Sudanese Animal Populations (BIOPASS) – Research Instititue for Development (IRD), Bel0Air PO Box 1386, Dakar, Senegal

Summary of the research:

The authors investigated several different maize varieties for resistance to the larger grain borer. They looked at weight loss, grain damage, number of emerged insects, median development time and susceptibility index. They did this using a no-choice test, where corn kernels of each variety were artificially infested and then evaluated for the previously mentioned parameters. They also looked to link physiochemical characteristics of the grain to varietal susceptibility by looking at grain moisture, total phenolics, palmitic acid, proteins, amylose, density, and grain hardness. The significance of this work is that farmers will have a better understanding of the varieties available and their ability to provide protection of the crop even after harvest. The study was able to identify 3 resistant varieties for potential use.

Methods:

Maize Varieties Used

Evaluated the varieties: Early-Thai, SWAN, DMR-ES, Tzee-White and Tzee-Yellow, Obatampa, Across-pool, Synth-C and Synth-9243.

Physiochemical Analysis

Moisture content was determined by drying, protein content was determined through nitrogen determination by Dumas methodology, starch content was determined using Ewers polarimetric method, amylose/amylopectin ratio of starches was evaluated with Morrison and Laignelet method, crude oil content was extracted via Folch et al., methodology, fatty acid content was determined after derivatization in BF3/Methanol mixture by gas chromatography, and total phenolics was also analyzed. They also evaluated grain hardness using the Stenvert hardness test, kernel density was determined from the ratio of grain weight to volume measured using Fox and Manley method. They had 3 replicates for each analysis.

Screening of insect-maize varieties interactions

Varietal susceptibility was assessed through no-choice artificial infestation. Three Male/Female pairs were placed into the jars of maize and the adults were removed after 14 days. The jars of infested kernels were kept in an insect chamber until 20 days post infestation. Emerged adults were then counted weekly until 55 days post infestation. They had 3 replicates for each maize variety.

Sample Size:

The authors made a point to emphasis the fact that they replicated their results at least 3 times for each important step.

Analysis seemed appropriate:

Analysis appears to be appropriate for the work presented. The methodology appears sound. At every step the authors made a point to site previous work which would back up their methodology. This is a very basic and already proven experimental design as far as testing for resistant varieties of grain.

Comments:

-Lines 16-36 The abstract seems too long and lacking a good introductory statement to pull the reader in, and a similarly strong concluding statement: Done

I would not put all these results and numbers in the abstract, give brief and clear statements about the findings. I would like to see these added, as many folks read the abstract first to decide: Done

1. What the paper is about and

2. Will they read all of it. 

-I was happy to see the results clearly separated from the discussion, which made it easier to evaluate.

-Rationale for the work was strong!

-I like the fact that the authors provide citations to strengthen their methods at every step and show that their methods have a strong background in the literature.

-I was pleased to see the authors went variety by variety and explained what they found in their analysis in detail in the discussion.

-Overall, considering the current literature published on the resistant varieties of maize to the larger grain borer, I think this paper is a needed contribution to the current body literature on the subject. There seem to be very few studies that look resistant varieties of maize for this species and many people in general are quite biased against host plant resistance in stored products. However, considering invasive and devastating pests like the larger grain borer, this strategy could be quite useful.

---

## [Editor Report · Decision Letter 1]

9 Apr 2020

Varietal susceptibility of maize to larger grain borer, Prostephanus truncatus (Horn) (Coleoptera : Bostrichidae), based on grain physicochemical parameters

PONE-D-19-25206R1

Dear Dr. NGOM,

We are pleased to inform you that your manuscript has been judged scientifically suitable for publication and will be formally accepted for publication once it complies with all outstanding technical requirements.

With kind regards,

Craig Eliot Coleman, PhD

Academic Editor

PLOS ONE
---

## [Editor Report · Acceptance letter]

13 Apr 2020

PONE-D-19-25206R1 

Varietal susceptibility of maize to larger grain borer, *Prostephanus truncatus* (Horn) (Coleoptera; Bostrichidae), based on grain physicochemical parameters      

Dear Dr. Ngom:

I am pleased to inform you that your manuscript has been deemed suitable for publication in PLOS ONE. Congratulations! Your manuscript is now with our production department. 

With kind regards,

on behalf of

Dr. Craig Eliot Coleman 

Academic Editor

PLOS ONE